# The utility of urinary biomarker panel in predicting renal pathology and treatment response in Chinese lupus nephritis patients

Li Liu[1☯], Ran Wang[2☯], Huihua Ding[2], Lei Tian[1], Ting Gao[1], Chunde Bao[2]*

1 Department of Emergency Medicine, Renji Hospital, Shanghai Jiao Tong University School of Medicine, Shanghai, China, 2 Department of Rheumatology, Renji Hospital, Shanghai Jiao Tong University School of Medicine, Shanghai, China

☯ These authors contributed equally to this work.
* baochunde_1678@126.com

**Data Availability Statement:** All relevant data are within the manuscript and its Supporting Information files.

## Abstract

Given the urgent need for non-invasive biomarkers of LN, we aim to identify novel urinary biomarkers that facilitate diagnosis, assessment of disease activity and prediction of treatment response in a retrospective SLE cohort. A total of 154 SLE patients and 55 healthy controls were enrolled, among whom 73 were active LN patients. We measured renal activity by renal SLEDAI. The treatment response of the active LN patients who finished 6-month induction therapy was assessed based on the American College of Rheumatology response criteria. The expression levels of 10 urinary biomarkers (UBMs): β2-MG, calbindin D, cystatin C, IL-18, KIM-1, MCP-1, nephrin, NGAL, VCAM-1, and VDBP were tested using Luminex high-throughput proteomics technology. All but urinary nephrin levels were significantly increased in active LN compared to healthy controls. uCystatinC, uMCP-1, uKIM-1 levels were significantly higher in active LN group compared to inactive LN group. Correlation analysis revealed positive correlation between uCystatinC, uKIM-1, uMCP-1, uNGAL, uVDBP and RSLEDAI score. In renal pathology, uCystatinC, uKIM-1, uVCAM-1, and uVDBP positively correlated with activity index (AI) while uVCAM-1 positively correlated with chronicity index (CI). Moreover, the combination of uVCAM-1, uCystatinC, uKIM-1 discriminated proliferative LN from membranous LN with an AUC of 0.80 (95%CI: 0.69–0.90). Most importantly, baseline uNGAL demonstrated good prediction ability to discriminate responders from non-responders in active LN patients after 6-month induction therapy. Using a multiplex bead technique, we have identified the combination of uVCAM-1, uCystatinC, uKIM-1 as a biomarker panel to reflect renal pathology and NGAL as a promising urinary biomarker to both reflect disease activity and predict treatment response.

## Introduction

Systemic lupus erythematosus (SLE) is a complex systemic autoimmune disease with unknown etiology. Lupus nephritis (LN) is one of most common organ involvements, affecting more than 60% of SLE patients during their disease courses [1, 2]. Despite aggressive

**Funding:** The study was supported by the National Natural Science Foundation of China (Grant no.: 81771733) received by CB. The funders had no role in study design, data collection and analysis, decision to publish, or preparation of the manuscript.

**Competing interests:** The authors have declared that no competing interests exist.

**Abbreviations:** SLE, Systemic lupus erythematosus; LN, Lupus nephritis; MCP-1, Monocyte chemoattractant protein-1; TWEAK, TNF-like weak ind 1 ucer of apoptosis; NGAL, Neutrophil gelatinase-associated lipocalin; VCAM, Vascular cell adhesion molecule-1; UBMs, Urinary biomarkers; β2-MG, Beta-2 microglobulin; KIM-1, Kidney injury molecule 1; VDBP, Vitamin D-binding protein; ACR, American College of Rheumatology; SLEDAI-2k, SLE Disease Activity Index 2000; RSLEDAI, Renal SLEDAI; ISN/RPS, International Society of Nephrology/Renal Pathological Society; AI, Activity index; CI, Chronicity index; PR, Partial response; CR, Complete response; NR, Nonresponse; IQR, Interquartile ranges; ANOVA, One-way analysis of variance; ROC, Receiver operating characteristic; AUC, Area under the curve; PPV, Positive predictive value; NPV, Negative predictive value; RAIL, Renal Activity in Lupus.

immunosuppression therapy, LN is still the major predictor of prognosis and an important driver of morbidity and mortality in SLE [3]. Moreover, SLE patients with glomerulonephritis have a significantly decreased health related life quality as well as working ability [4]. Although renal biopsy remains the gold standards of making the diagnosis, guiding the treatment, and predicting prognosis in LN patients, the invasive nature and associated risks have limited its use, especially in the follow-up stage [5]. On the other hand, conventional biomarkers, like proteinuria, anti-dsDNA antibody and complement, are neither sensitive nor specific in predicting renal activity in LN patients [6]. There is an urgent need for non-invasive biomarkers to reflect renal activity, predict renal prognosis and ultimately guide the treatment of LN.

The past decade has witnessed a notable progress in the identification and validation of biomarkers of LN. An increasing number of researchers have focused on identifying biomarkers of LN from urine due to its availability through noninvasive collection and direct reflection of what's going on in the kidney. With the development of high-throughput technology, the strategy of biomarker discovery has changed. Previous studies have identified a variety of urinary markers of LN in both adult and pediatric patients through different strategies, including monocyte chemoattractant protein-1 (MCP-1) [7, 8], TNF-like weak inducer of apoptosis (TWEAK) [9], neutrophil gelatinase-associated lipocalin (NGAL) [7, 8], and vascular cell adhesion molecule-1 (VCAM) [7, 10].

Although the previous described urinary biomarkers (UBMs) has been validated widely, it is believed that a single biomarker may not be sufficient for clinical application in lupus nephritis. Previous validation studies on LN urine biomarkers largely focused on limited number (usually less than five) of markers. In this study, we aim to use a more sensitive multiplex technology to simultaneously quantitate ten potential UBMs and establish the potential role of urine protein biomarker panel in reflecting disease activity and predicting treatment response in LN patients. The ten proteins we selected are beta-2 microglobulin (β2-MG), calbindin D, cystatin C, IL-18, kidney injury molecule 1 (KIM-1), monocyte chemoattractant protein 1 (MCP-1), nephrin, neutrophil gelatinase-associated lipocalin (NGAL), vascular cell adhesion molecule 1 (VCAM-1), and vitamin D-binding protein (VDBP).

## Materials and methods

### Study population

In this retrospective cohort study, a total of 154 SLE patients (94% women) above 18 years of age from Renji Hospital, Shanghai Jiao Tong University School of Medicine were recruited from August 1, 2016 to August 1, 2018, including 73 active LN, 32 inactive LN, 49 non-renal SLE. 55 age and gender matched healthy individuals were recruited as controls. All patients fulfilled the 1997 American College of Rheumatology (ACR) classification criteria for SLE [11]. The diagnosis of LN was confirmed by renal biopsy. All patients with drug induced lupus, active malignancies, overlapping syndromes, end-stage renal disease, urinary tract infection, active systemic infection and history of renal transplantation were excluded from the study. The 73 active LN patients received induction therapy according to their physicians and were followed up for 6 months after induction therapy.

This study was approved by the ethics committee of Renji Hospital, Shanghai Jiao Tong University School of Medicine and conducted in accordance with good clinical practice. All patients have provided written informed consent.

### Data collection

Demographic and clinical information including age, sex and clinical manifestations were collected through chart review by an experienced rheumatologist. Assays of complete blood

count, serum creatinine, eGFR, dsDNA, C3, C4, autoantibodies, and 24-hour urine protein levels were performed as routine laboratory tests and data were collected from electronic records. Corticosteroids and immunosuppressive agents use information were collected in SLE patients. SLE Disease Activity Index 2000 (SLEDAI-2k) and renal SLEDAI (RSLEDAI, the total score of the four kidney-related parameters in SLEDAI-2k) were calculated based on chart review according to literature [12]. Authors had access to information that could identify individual participants during data collection.

## Disease assessment and follow-up

At the time of enrollment, we used SLEDAI-2k and RSLEDAI to evaluate global disease activity and renal disease activity respectively. Renal biopsies were reviewed and classified by an experienced renal pathologist based on the 2003 International Society of Nephrology/Renal Pathological Society (ISN/RPS) classification [13]. Renal histological activity was assessed by activity index (AI) and chronicity index (CI) as described elsewhere [14]. According to the renal histology, patients with class III, IV, class III plus V, or class IV plus V LN were defined as proliferative LN, patients with pure class V LN were defined as membranous LN. Active LN were defined as proteinuria of more than 0.5g/24h at the time of enrollment. Inactive LN was defined as RSLEDAI = 0 at the time of enrollment. Detailed demographic and clinical characteristics of different groups of SLE patients were listed in Table 1.

The treatment response of the patients who finished 6-month induction therapy was assessed based on American college of rheumatology response criteria for proliferative and membranous renal disease in systemic lupus erythematosus clinical trials [15] as follows. Complete response (CR): eGFR of >90 ml/minute/1.73m$^2$, urinary protein creatinine ratio (UPCR) <0.2. Partial response (PR): no more than 25% decline in estimated GFR or end-stage renal disease, at least 50% reduction in the UPCR and UPCR is 0.2–2.0. Non-response (NR): not meeting the CR or PR criteria. Based on eGFR and UPCR, the patients were divided into CR, PR and NR groups. We didn't include urine sediment in the criteria of treatment response due to the poor reproductivity.

## Assay of urinary protein markers

Midstream morning urine was collected from each participant in a sterile container using an aseptic technique, which was immediately centrifuged after collection. The supernatant was aliquoted and stored at -80˚C refrigerator until analysis. The expression levels of β2-MG, calbindin D, cystatin C, IL-18, KIM-1, MCP-1, nephrin, NGAL, VCAM-1, and VDBP were tested using Luminex high-throughput proteomics technology (kits from R&D LXSAHM-11 and LXSAHM-02) according to the manufacturer's instructions. Urinary protein levels were normalized by urinary creatinine to correct the effect of urine concentration.

## Statistical analyses

Statistical analysis was performed using SPSS21 and Prism7.01. Continuous variables were expressed as mean ± standard deviation for normally distributed variables or median and interquartile ranges (IQR), otherwise. Normality of data was established by Shapiro—Wilk tests. Student's t test was used to compare the means of continuous variables conforming to normal distribution between groups. Mann-Whitney U test was used for continuous variables non-conforming to the normal distribution. One-way analysis of variance (ANOVA) was used to compare three or more groups of data with normal distribution while Kruskal-Wallis H test for non-normal distribution data. Dichotomous variables were expressed as counts and percentages and comparison between groups was performed by Chi-square test. Correlation

**Table 1. Demographic and clinical characteristics of SLE patients.**

| | Active LN | Inactive LN | Non-renal SLE |
|---|---|---|---|
| | (n = 73) | (n = 32) | (n = 49) |
| Sex (F/M) n | 69/4 | 31/1 | 45/4 |
| Age (years) Mean±SD | 35.71±14.08 | 33.22±10.20 | 35.90±11.67 |
| **Clinical Manifestations** | | | |
| Fever n(%) | 4(5.48) | 3(9.38) | 4(8.16) |
| Rash n(%) | 17(23.29) | 3(9.38) | 9(18.37) |
| Vasculitis n(%) | 1(1.37) | 0 | 1(2.04) |
| Ulcer n(%) | 12(16.44) | 2(6.25) | 6(12.24) |
| Serositis n(%) | 7(9.59) | 2(6.25) | 3(6.12) |
| Arthritis n(%) | 15(20.55) | 3(9.38) | 3(6.12) |
| NPSLE n(%) | 0 | 0 | 0 |
| Nephritis n(%) | 73(100) | 0 | 0 |
| PAH n(%) | 0 | 0 | 0 |
| Hematologic n(%) | 8(10.96) | 1(3.13) | 6(12.24) |
| **Laboratory test** | | | |
| WBC (10^9/L) Median(IQR) | 6.89 (4.57–9.63) | 5.15 (4.38–6.58) | 5.00 (4.20–6.69) |
| Hb (g/L) Mean±SD | 107.99±23.23 | 126.09±26.56 | 122.67±20.66 |
| Plt (10^9/L) Median (IQR) | 199 (153–250) | 224 (170–264) | 204 (131–227) |
| ESR (mm/h) Median (IQR) | 29(13–49) | 14(7–29) | 13(9–36) |
| CRP (mg/L) Median (IQR) | 3.13(3.00–4.65) | 1.27(0.61–3.66) | 1.90(0.78–3.28) |
| MDRD-GFR (mL/(min*1.73m2)) Mean±SD | 97.63±42.80 | 119.98±31.34 | 142.63±172.84 |
| Serum Creatinine (μmol/L) Median (IQR) | 65 (54–89) | 54 (50–63) | 55 (49–64) |
| 24h urine protein (g/24h) Median (IQR) | 2.42 (1.32–4.76) | - | - |
| Complement C3 (g/L) Mean±SD | 0.62±0.26 | 0.88±0.20 | 0.79±0.25 |
| Complement C4 (g/L) Median (IQR) | 0.08 (0.06–0.14) | 0.15 (0.13–0.20) | 0.13 (0.10–0.18) |
| dsDNA (IU/ml) Median (IQR) | 63.36 (13.57–100.00) | 14.78 (8.92–23.55) | 27.13 (10.10–82.92) |
| **Autoantibodies, n of positive subjects (%)** | | | |
| Anti-Sm | 7 (9.5) | 2 (6.3) | 6 (12.2) |
| Anti-SSA/Ro | 39 (53.4) | 14 (43.8) | 26 (53) |
| Anti-RNP | 22 (30.1) | 10 (31.3) | 11 (22.4) |
| Anti-SSB | 8 (11.0) | 2 (6.3) | 6 (12.2) |
| Anti- Nucleosome | 25 (34.2) | 1 (3.1) | 7 (14.3) |
| Anti- Ribosomal -P | 13 (17.8) | 1 (3.1) | 5 (10.2) |
| Anti- Histone | 9 (12.3) | 0 | 4 (8.2) |
| APL | 1 (1.4) | 1 (3.1) | 1 (2.0) |
| SLEDAI, Median (IQR) | 8 (8–12) | 2 (0–3) | 2 (2–4) |
| RSLEDAI, Median (IQR) | 4 (4–8) | 0 (0) | 0 (0) |
| **Medications** | | | |
| Pred (mg) Median (IQR) | 30 (20–50) | 7.5 (2.5–11.5) | 7.5 (2.5–15) |
| Methotrexate n (%) | 0 (0) | 1 (3.1) | 0 (0) |
| Azathioprine n (%) | 1(1.4) | 7 (21.9) | 3 (6.1) |
| CsA n (%) | 9 (12.3) | 4 (12.5) | 2 (4.1) |
| Tacrolimus n (%) | 8 (11.0) | 3 (9.4) | 0 (0) |
| Leflunomide n (%) | 1 (1.4) | 6 (18.8) | 7 (14.3) |
| MMF n (%) | 10 (13.7) | 7 (21.9) | 1 (2.0) |
| CYC n (%) | 38 (52.1) | 3 (9.4) | 0 (0) |
| Iguratimod n (%) | 5 (6.8) | 2 (6.3) | 1 (2.0) |

*(Continued)*

**Table 1.** (Continued)

|  | Active LN | Inactive LN | Non-renal SLE |
|---|---|---|---|
|  | (n = 73) | (n = 32) | (n = 49) |
| Thalidomide n (%) | 1(1.4) | 0 (0) | 0 (0) |
| **Histological type** |  |  |  |
| Proliferative LN n (%) | 54 (74.0) |  |  |
| Class III, IV, III+V, IV+V n | 6, 24, 11, 13 |  |  |
| AI Median (IQR) | 7 (4–9) |  |  |
| CI Median (IQR) | 4 (3–6) |  |  |
| Membranouse LN n (%) | 19 (26.0) |  |  |
| Class V n | 19 |  |  |
| AI Median (IQR) | 1 (0–3) |  |  |
| CI Median (IQR) | 3 (2–5) |  |  |

LN, lupus nephritis; SLE, systemic lupus erythematosus; SD, standard deviation; F, female; M, male; NPSLE, Neuropsychiatric systemic lupus erythematosus; PAH, pulmonary arterial hypertension; WBC, white blood cell; Hb, hemoglobin; Plt, platelet; ESR, erythrocyte sedimentation rate; CRP, C-reactive protein; MDRD, modification of diet in renal disease Study; GFR, glomerular filtration rate; dsDNA, anti-double-stranded DNA antibody; anti-Sm, anti-Smith; Anti-SSA/Ro, Anti-Sjögren's-syndrome-related antigen A/Ro; anti-RNP anti-ribonucleoprotein; Anti-SSB, anti-Sj gren syndrome B; APL, anti-phorpholipid; SLEDAI, SLE disease activity index; RSLEDAI, renal SLEDAI; AI, Activity Index; CI, Chronicity Index; Pred: prednisone; CsA: cyclosporin A; MMF: mycophenolate mofetil; CYC: cyclophosphamide; CR: complete response; PR: partial response; NR: Non-response.

analysis was performed using Spearman's rank correlation. Receiver operating characteristic (ROC) curve analysis was employed to evaluate the diagnostic value of various biomarkers in differentiating active LN from inactive LN. The area under the curve (AUC) was calculated and the best trade-off point of sensitivity and specificity was determined from the values calculated for each of the coordinates on the curve. A two-tailed value of $p < 0.05$ was considered as statistically significant.

## Results

A total of 154 SLE patients (94% female) were enrolled in this study, including 73 active LN (47%), 32 inactive LN (21%), 49 non-renal SLE (32%). 55 age (34.35±7.49) and gender (90% female) matched healthy individuals were recruited as controls. According to the renal pathology results, active LN patients were divided into proliferative LN group (54 cases) and membranous LN group (19 cases). The 73 active LN patients received induction therapy with prednisone plus mycophenolate mofetil (n = 10), cyclophosphamide (n = 38), cyclosporine A (n = 9), tacrolimus (n = 8), azathioprine (n = 1), leflunomide (n = 1), iguratimod (n = 5), thalidomide (n = 1). After 6 months of induction therapy, 17 patients were lost to follow up. In the 56 active LN patients who finished 6 months induction therapy and follow-up, 17 achieved complete response, 27 achieved partial response while the other 12 were non-responders (S1 Fig).

### The comparison of UBMs levels in different groups

Compared with healthy controls, all UBMs except nephrin were significantly increased in the active LN group (Table 2). Besides, uCystatinC, uMCP-1, uKIM-1, and uVDBP levels were significantly elevated in active LN patients compared to those in SLE patients without renal involvement (Table 2). More importantly, uCystatinC, uMCP-1, uKIM-1 levels were significantly higher in active LN group compared to inactive LN group (Table 2, Fig 1A–1C).

**Table 2. UBMs levels in different groups of patients[a].**

| | *Active LN* | *Inactive LN* | *adj p[b]* | *NonLN SLE* | *adj p[b]* | *Healthy Control* | *adj p[b]* |
|---|---|---|---|---|---|---|---|
| *β2-MG* | 2161.691 (829.994–3175.458) | 1716.589 (598.933–2659.219) | | 1204.276 (567.577–2569.795) | | 657.147 (37.135–1423.217) | *** |
| *Calbindin D* | 12.675 (6.448–21.341) | 7.843 (4.821–16.78) | | 7.133 (4.017–11.763) | | 4.979 (1.78–11.857) | *** |
| *Cystatin C* | 930.812 (356.528–2764.948) | 350.861 (205.758–686.961) | * | 467.373 (194.697–878.543) | * | 340.074 (21.04–916.834) | *** |
| *IL-18* | 0.183 (0.098–0.637) | 0.269 (0.124–0.425) | | 0.195 (0.077–0.465) | | 0.07 (0.032–0.174) | *** |
| *KIM-1* | 19.326 (8.309–59.182) | 5.881 (2.41–10.258) | ** | 3.658 (1.52–12.003) | *** | 0.688 (0.147–6.739) | *** |
| *MCP-1* | 2.606 (0.87–8.549) | 1.064 (0.442–1.882) | * | 0.795 (0.283–1.911) | ** | 0.322 (0.104–1.066) | *** |
| *Nephrin* | 11.011 (2.755–31.34) | 8.67 (4.931–17.399) | | 6.443 (2.452–19.241) | | 4.382 (1.84–15.303) | |
| *NGAL* | 1445.442 (713.418–5380.318) | 998.742 (436.091–1784.045) | | 585.126 (275.972–1359.18) | | 1456.635 (471.465–6162.635) | ** |
| *VCAM-1* | 914.258 (253.363–2900.686) | 858.611 (367.303–1212.107) | | 586.904 (317.004–1560.023) | | 237 (104.6–1876.378) | * |
| *VDBP* | 407.814 (91.056–2705.239) | 111.505 (71.744–247.134) | | 49.342 (14.092–104.59) | *** | 17.941 (10.325–32.648) | *** |

a. Data were expressed as median (IQR) since they were not normally distributed, unit: pg/mmol*Creatinine.

b. adj p: P value versus active LN group after bonferroni correction.

β2-MG: beta-2 microglobulin; MCP-1: monocyte chemoattractant protein 1; VDBP: vitamin D-binding protein; NGAL: neutrophil gelatinase-associated lipocalin;

KIM-1: kidney injury molecule 1; VCAM-1: vascular cell adhesion molecule 1

* $p < 0.05$

** $p < 0.01$

*** $p < 0.001$.

To assess the capability of uCystatinC, uMCP-1, uKIM-1 to discriminate active LN from inactive LN, we performed ROC curve analysis. The AUC value was 0.69 (95% CI: 0.59–0.79), 0.70 (95% CI: 0.60–0.80), 0.76 (95% CI: 0.66–0.86) for uCystatin C, uMCP-1, uKIM-1 separately (Fig 1D).

## The correlation between UBMs and clinical parameters

The correlation analysis revealed that uCystatinC, uKIM-1, uMCP-1, uNGAL, and uVDBP positively correlated with the total amount of 24-hour urinary protein. Uβ2-MG, uCalbindinD, uCystatinC, uKIM-1, uMCP-1, uNephrin, uNGAL, and uVDBP positively correlated with SLEDAI score while uCystatinC, uKIM-1, uMCP-1, uNGAL, and uVDBP positively correlated with RSLEDAI score. More importantly, uCystatinC, uKIM-1, uVCAM-1, and uVDBP positively correlated with AI while uVCAM-1 positively correlated with CI in renal pathology (Table 3).

## UBMs as potential markers of renal pathology

With the concurrent renal biopsy, we were able to compare the levels of UBMs between proliferative LN and membranous LN. In this study, proliferative LN (n = 54) referred to class III, class IV, mixed class III/V, and mixed class IV/V while membranous LN referred to pure class V (n = 19). The study showed significantly increased uCystatinC, uKIM-1, and uVCAM-1 in proliferative LN when compared to those in membranous group (Fig 2A–2C). ROC analysis revealed an AUC of 0.76 (95% CI: 0.648–0.873), 0.69 (95% CI: 0.56–0.81), 0.67 (0.54–0.80), and 0.80 (95%CI: 0.69–0.90) for uVCAM-1, uCystatinC, uKIM-1 and the combined three of the above UBMs. When using the combined three UBMs to discriminate membranous LN from proliferative LN, the sensitivity and specificity were 94.7% and 55.6%, the positive predictive value (PPV) and negative predictive value (NPV) were 42.9% and 96.8%. (Fig 2D).

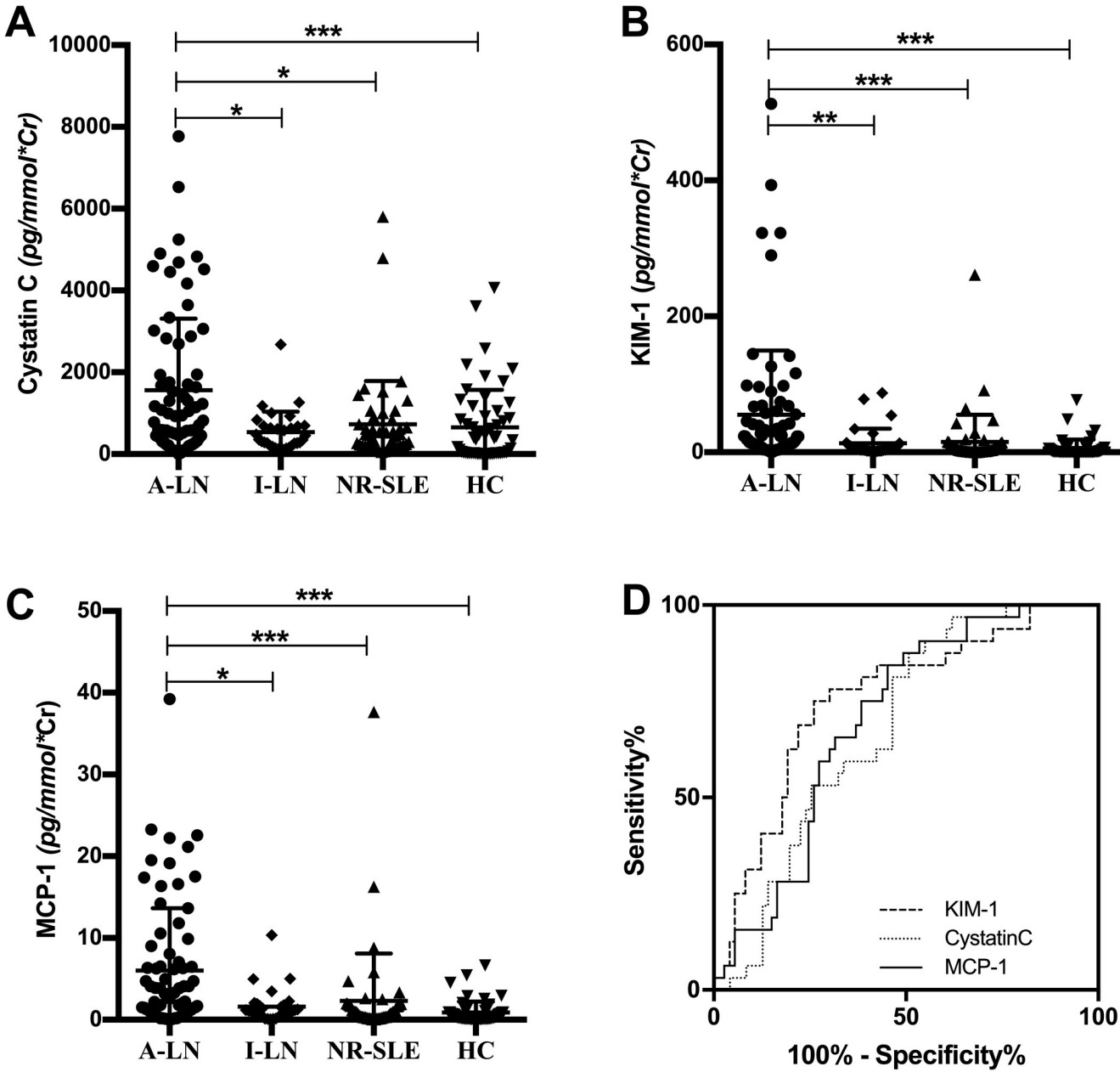

**Fig 1. Comparison of three urinary markers in LN patients and healthy controls.** Comparison of uCystatinC (A), uKIM-1(B), and uMCP-1(C) levels in four groups of patients showed significant increase of the urinary markers in active LN patients. ROC curve for the three UBMs showed good performance to differentiate active LN from inactive LN (D). A-LN: active lupus nephritis, I-LN: inactive lupus nephritis, NR-SLE: nonrenal SLE, HC: healthy controls, KIM-1: kidney injury molecule 1, MCP-1: monocyte chemoattractant protein 1.

### NGAL as a potential marker of treatment response

In the subgroup of active LN patients who finished 6-month induction therapy, we further explore the difference of baseline UBMs levels in patients with different treatment responses. After 6 months of induction therapy, 17 patients were lost to follow up. In the 56 active LN patients who finished 6 months induction therapy and follow-up, 17 achieved complete response, 27 achieved partial response while the other 12 were non-responders. Baseline

**Table 3. Correlation of UBMs with disease activity scores.**

|  | 24hUP | | SLEDAI-2k | | RSLEDAI | | AI | | CI | |
|---|---|---|---|---|---|---|---|---|---|---|
|  | Spearmen r | p | Spearmen r | p | Spearmen r | p | Spearmen r | p | Spearmen r | p |
| β2-MG | 0.07333 |  | 0.2293 | ** | 0.04035 |  | -0.05865 |  | 0.04482 |  |
| Calbindin D | 0.1066 |  | 0.2262 | ** | 0.04808 |  | -0.05053 |  | -0.04973 |  |
| Cystatin C | 0.3135 | *** | 0.3586 | *** | 0.3228 | *** | 0.3759 | * | 0.2782 |  |
| IL-18 | 0.09846 |  | 0.1352 |  | -0.1598 |  | 0.02824 |  | 0.04967 |  |
| KIM-1 | 0.4292 | *** | 0.456 | *** | 0.4875 | *** | 0.3601 | * | -0.00195 |  |
| MCP-1 | 0.34 | *** | 0.4116 | *** | 0.3471 | *** | 0.2203 |  | 0.09216 |  |
| Nephrin | 0.141 |  | 0.1916 | * | -0.04185 |  | -0.03963 |  | 0.02619 |  |
| NGAL | 0.2592 | ** | 0.3702 | *** | 0.4183 | *** | -0.02648 |  | -0.1411 |  |
| VCAM-1 | 0.1463 |  | 0.1111 |  | 0.05509 |  | 0.3686 | * | 0.3014 | * |
| VDBP | 0.4332 | *** | 0.4165 | *** | 0.3231 | ** | 0.3526 | * | -0.03098 |  |

24hUP: 24-hour urine protein; SLEDAI-2k: SLE Disease Activity Index 2000; RSLEDAI: renal SLEDAI; AI: activity index; CI: chronicity index; β2-MG: beta-2 microglobulin; MCP-1: monocyte chemoattractant protein 1; VDBP: vitamin D-binding protein; NGAL: neutrophil gelatinase-associated lipocalin; KIM-1: kidney injury molecule 1; VCAM-1: vascular cell adhesion molecule 1

\* p<0.05

\*\*p<0.01

\*\*\*p<0.001.

characteristics of patients with different response status were listed in S1 Table. Among the 10 UBMs, only baseline uNGAL levels were significantly lower among those with complete response (553.68 ng/mL; 95%CI 306.58–1696.13) than those in patients with partial response (1073.44 ng/mL; 95% CI 240.22–1883.89) and nonresponse (2621.25ng/mL; 95% CI 2052.78–3147.47) (Fig 3A). No significant differences were observed in the other 9 urinary biomarkers after treatment. To determine the performance of uNGAL to predict renal response, we performed ROC analysis. The AUC for uNGAL to discriminate responders from non-responders after 6-month induction therapy was 0.78 (95% CI: 0.65–0.92). At the cut-off value of 1964.58 ng/mL, the sensitivity was 81.4%, specificity was 83.3%, PPV was 55.6%, and NPV was 94.6% (Fig 3B).

## Discussion

Renal involvement is a common and serious complication of SLE, making it an important predictor of survival in SLE patients [1]. Despite advances in the treatment of lupus nephritis, its management is fraught with uncertainty and lack of reliable biomarkers for intrarenal activity and chronicity. In this study, we have validated 10 potential urinary biomarkers, which were previously mentioned to be promising candidate biomarkers of LN, using a multiplex assay. Our results indicated that in out cohort of Chinese LN patients, using a panel of urinary biomarkers, we were able to validate that the combination of uVCAM-1, uCystatin C, uKIM-1 could be a promising biomarker panel for discriminate proliferative LN from membranous LN. More importantly, using a longitudinal cohort of LN, who received 6-month induction therapy after renal biopsy, we have validated increased baseline uNGAL as a good predictor for treatment response.

The biomarkers panel in this study was selected based on previous studies showing potential of being candidate biomarkers for either LN or other renal diseases [16–26]. Not surprisingly, the results of the current study further corroborate findings of the previous LN biomarker studies. Urinary markers with the most evidence in LN, such as MCP-1, VCAM-1,

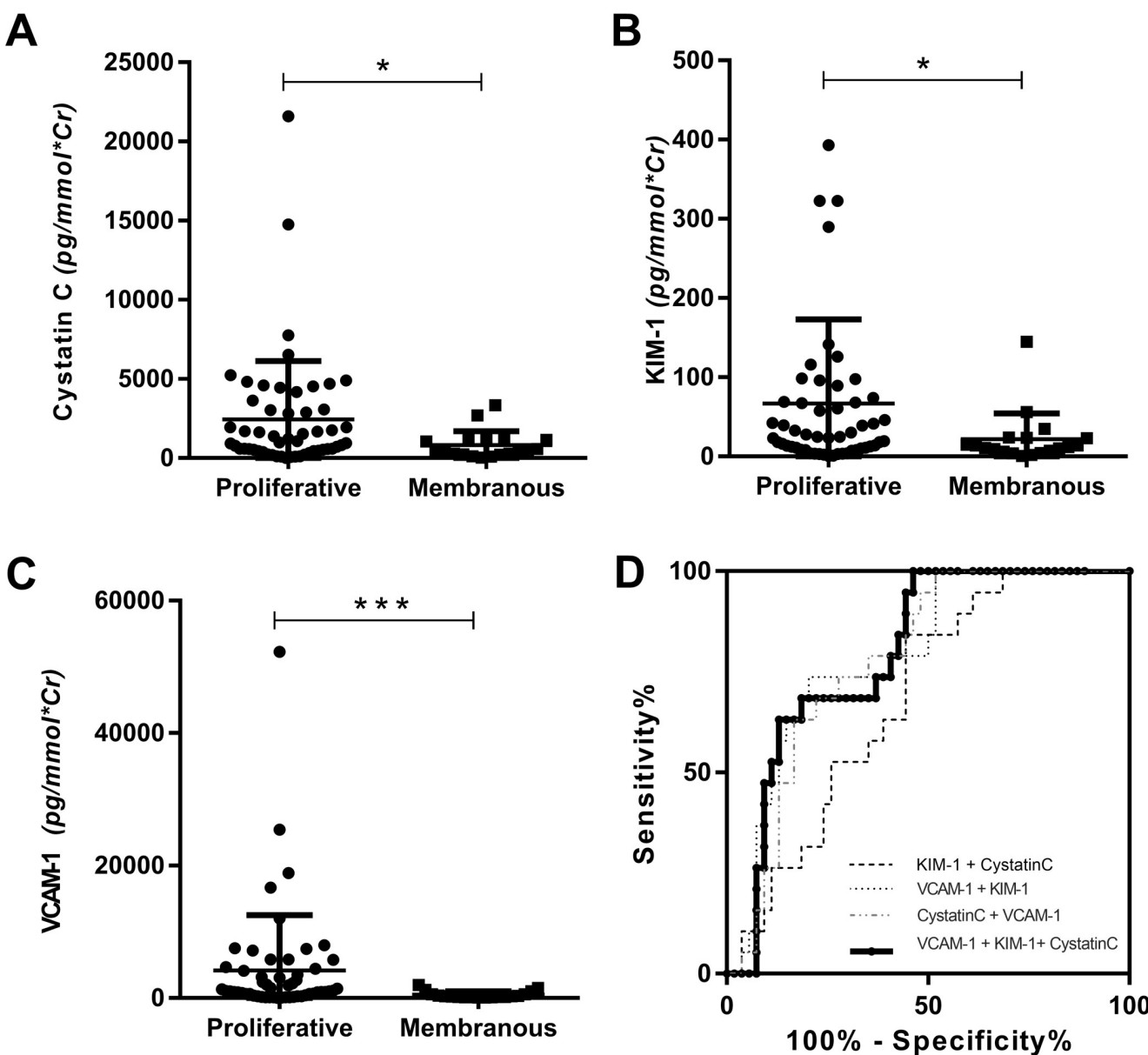

**Fig 2. Comparison of three urinary markers in different pathologic classes of lupus nephritis.** Comparison of uCystatinC (A), uKIM-1(B), and uVCAM-1(C) levels in proliferative and membranous LN patients and ROC curve for the combined two and three UBMs to differentiate proliferative LN from membranous LN (D). KIM-1: kidney injury molecule 1, VCAM-1: vascular cell adhesion molecule 1, Sen: sensitivity, Spe: specificity, PPV: positive predictive value, NPV: negative predictive value, AUC: area under the curve.

and NGAL, were proved to perform better as diagnostic or prognostic markers in this study. In this study, uMCP-1 levels were significantly higher in active LN patients when compared to inactive LN, non-renal SLE and healthy controls, which is in accordance with the result in a recent meta-analysis study [26]. The correlation ship between uMCP-1 and renal disease activity [27, 28] was also confirmed in the current study.

uVCAM-1 was another widely validated biomarker of LN [21, 23, 25, 29, 30]. Our study confirmed elevated uVCAM-1 in active LN patients when compared to healthy control. However, in contrast to the earlier findings, we didn't observe any difference in uVCAM-1 level

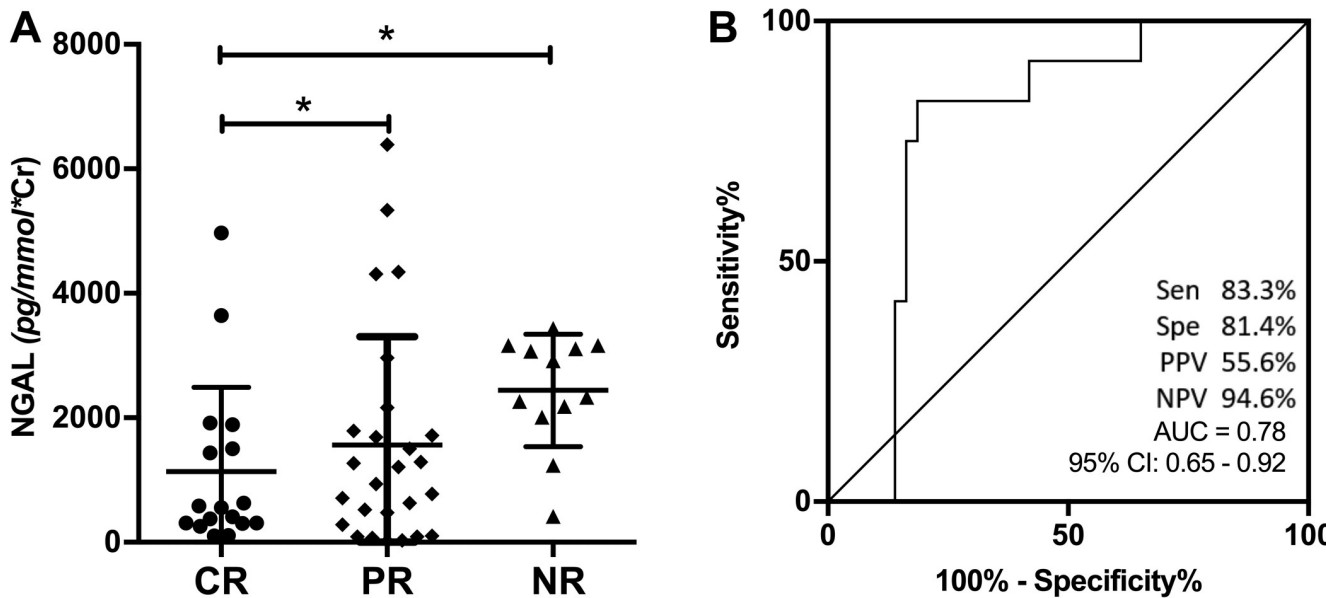

**Fig 3. NGAL as a potential marker of treatment response.** Comparison of NGAL in LN patients with different response status after 6 months of treatment (A) and ROC curve for NGAL to differentiate responders from non-responders (B). NGAL: neutrophil gelatinase-associated lipocalin, CR: complete response, PR: partial response, NR: Non- response, Sen: sensitivity, Spe: specificity, PPV: positive predictive value, NPV: negative predictive value, AUC: area under the curve.

between active LN group and inactive LN group. Neither did we establish any correlation between uVCAM-1 and disease activity. This might be due to the different study population. The majority of patients studied in earlier publications were not Asian ethnicity while our study focused in Chinese LN patients. Another possible explanation for this is that the assay used for uVCAM-1 in our study was a multiplex bead technology, which is more sensitive in terms of detection range, while previous studies mostly used enzyme-linked immunosorbent assay (ELISA). In our study, uVCAM-1 correlated positively with both AI and CI in renal pathology, indicating the potential role of uVCAM-1 in reflecting renal pathology changes. We further observed that the combination of uVCAM-1, uCystatin C, uKIM-1 can discriminate membranous LN from proliferative LN. This finding is in accordance with a recent study which confirmed the role of uVCAM-1 in predicting elevated renal pathology AI in LN [25].

Serum cystatin C was broadly reported as a biomarker for renal dysfunction in LN and other renal diseases [31, 32]. However, only one recent study discussing the effect of preservatives on urine protein degradation mentioned elevated urinary levels of cystatin C in LN patients [33]. We have demonstrated that uCystatin C levels were significantly increased in active LN and correlated with 24h urine protein, SLEDAI, RSLEDAI, and AI in renal pathology. To our knowledge, this is the first publication to describe the potential correlation between uCystatin C and disease activity in LN.

KIM-1 was first reported as a biomarker for acute kidney injury [34]. Urinary KIM-1 was reported to elevated in active LN patients and correlate with renal histological inflammation [35]. More recently, KIM-1 has been validated as an important component in several different urinary biomarker panels including RAIL for predicting renal pathology [36–38]. In the current study, we also demonstrated that when combined with other UBMs, KIM-1 was able to better reflect renal pathology.

uVDBP was only recently proved to be a biomarker for LN. A Korean group identified several urinary biomarkers using proteomics and validated uVDBP as a biomarker for reflecting

renal disease activity and predicting renal flare [16]. Our study further confirmed their results. We consider that in Asian LN patients, uVDBP can reflect urine protein level, renal disease activity and renal pathology AI.

The most important finding in this study is the predictive value of baseline uNGAL in treatment response after 6-month induction therapy. Urinary NGAL has been validated as a predictor of renal disease activity [19, 39], which has been confirmed in the current study. It is also one of the 6 UBMs in RAIL [36, 38] that predict renal pathology AI in LN patients. More recently, Satirapoja et al. demonstrated that baseline uNGAL performed better than conventional markers in predicting a treatment response in active LN patients with both sensitivity and specificity around 70% [20]. Although urinary NGAL level correlated well with 24h urine protein and predicted renal disease activity, it couldn't differentiate proliferative LN from membranous LN. Previous studies also haven't validated the correlation ship between urinary NGAL and the histopathological type of LN. Since patients' baseline character including renal pathological characteristics were not predictors for treatment response [40, 41], it's not surprising that urinary NGAL could predict treatment response but not renal histopathology type while the biomarkers which could differentiate different pathological types of LN could not predict treatment response.

Most of the biomarkers validated in this study, however, are not specific to LN. For instance, urinary KIM-1 and NGAL has been reported as biomarkers of kidney injury in obese children [42]. The current study didn't include patients with other chronic kidney diseases. We envision the clinical implication of the current study is to use these biomarkers in predicting renal pathology features or treatment response of LN after patients' diagnosis of SLE base on the classification criteria.

Given the complexity of the pathogenesis of lupus nephritis, it is now widely accepted that a single urinary biomarker might not be powerful enough. Many groups have tried different approaches to address this [36, 38, 43–45]. In this study, we used a multiplex technology to simultaneously measure 10 candidate UBMs. We have demonstrated that the technology is a suitable tool with wider analytic range and better cost-effectiveness. It is also proved to be stable in both urine and serum samples by others [46]. Using the multiplex technology, it is possible to validate hundreds of biomarkers with only limited volume of samples, which is extremely important when the sample is precious, for example, cerebrospinal fluid.

Our research, however, is subject to several limitations. First of all, the study population and sample size are limited. In particular, the longitudinal cohort with treatment responses was small. The results of the current study are valid with Chinese LN patients. The interpretation of the study conclusion in other patients' group should be with caution. Secondly, given the observational nature of the study, the induction treatment in the study population didn't strictly follow the international guidelines on the management of LN, in which 41/56 (73%) patients received first line treatment (CYC or MMF) recommended by LN guidelines. Due to the widely accepted evidence of calcineurin inhibitors in Asian LN patients, another 20% (11/56) patients received cyclosporin A or tacrolimus. The diversity of treatment regimens may bring indeed certain confounding bias in the study. In future plans, our study findings will need to be validated in larger prospective cohorts with more standardized treatment and better designed control groups.

## Conclusions

In conclusion, using a multiplex bead technique, we have identified a panel of urinary biomarkers (uVCAM-1, uCystatinC, uKIM-1) to reflect renal histology (proliferative LN vs membranous LN) in LN patients. Furthermore, urinary NGAL demonstrated good prediction

ability to discriminate responders from non-responders in active LN patients after 6-month induction therapy.

## Supporting information

**S1 Fig. Study flow diagram.**
(TIF)

**S1 Table. Baseline characteristics of patients with different renal responses.**
(DOCX)

## Author Contributions

**Conceptualization:** Li Liu, Ran Wang, Chunde Bao.

**Data curation:** Li Liu, Ran Wang, Huihua Ding, Lei Tian, Ting Gao.

**Formal analysis:** Li Liu, Ran Wang, Huihua Ding.

**Funding acquisition:** Chunde Bao.

**Investigation:** Li Liu, Ran Wang, Lei Tian.

**Methodology:** Li Liu, Ran Wang, Huihua Ding, Ting Gao, Chunde Bao.

**Project administration:** Li Liu, Huihua Ding, Chunde Bao.

**Resources:** Li Liu.

**Software:** Ran Wang, Huihua Ding, Lei Tian, Ting Gao.

**Supervision:** Huihua Ding, Chunde Bao.

**Validation:** Li Liu, Ran Wang.

**Visualization:** Li Liu, Ran Wang, Huihua Ding, Lei Tian, Ting Gao.

**Writing – original draft:** Li Liu, Ran Wang, Huihua Ding, Lei Tian, Ting Gao.

**Writing – review & editing:** Huihua Ding, Chunde Bao.

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
