## [Decision Letter · Decision Letter 0]

21 Jul 2020

PONE-D-20-15842

The utility of urinary biomarker panel in predicting renal pathology and treatment response in Chinese lupus nephritis patients

PLOS ONE

Dear Dr. Bao,

Thank you for submitting your manuscript to PLOS ONE. After careful consideration, we feel that it has merit but does not fully meet PLOS ONE’s publication criteria as it currently stands. Therefore, we invite you to submit a revised version of the manuscript that addresses the points raised during the review process.

Two reviewers found some interests in this study, but also pointed out a number of issues that require improvement or amendment. I ask the authors to fully respond to all comments made by reviewers in the revised version.

We look forward to receiving your revised manuscript.

Kind regards,

Masataka Kuwana, MD, PhD

Academic Editor

PLOS ONE

Journal Requirements:

2. In the ethics statement in the manuscript and in the online submission form, please provide additional information about the patient records used in your retrospective study. Specifically, please ensure that you have discussed whether all data were fully anonymized before you accessed them and/or whether the IRB or ethics committee waived the requirement for informed consent. If patients provided informed written consent to have data from their medical records used in research, please include this information.

"This study was supported by the National Natural Science Foundation of China grants (81771733)."

Reviewers' comments:

Reviewer's Responses to Questions

**Comments to the Author**

1. Is the manuscript technically sound, and do the data support the conclusions?

Reviewer #1: Yes

Reviewer #2: Partly

2. Has the statistical analysis been performed appropriately and rigorously? 

Reviewer #1: Yes

Reviewer #2: Yes

3. Have the authors made all data underlying the findings in their manuscript fully available?

Reviewer #1: Yes

Reviewer #2: No

4. Is the manuscript presented in an intelligible fashion and written in standard English?

Reviewer #1: Yes

Reviewer #2: Yes

5. Review Comments to the Author

Reviewer #1: Liu L et al investigated a large number of SLE patients to identify the predictor for renal pathological activity and prognosis by using 10-urinary biomarkers. This study is focusing on the important issue estimating activity of lupus nephritis without pathological confirmation. This manuscript sounds interesting, but I have concerns which authors should address.

Major

Urinary NGAL was highly correlated with both 24-hours proteinuria and RSLEDAI comparing to other urinary biomarkers (Table 3) and only the urinary biomarker predicting renal response. However, it could not distinguish proliferative LN from membranous LN while uVCAM-1, uCystatinC, and uKim-1 were significantly increased in proliferative LN (Figure 2). Author needs to discuss this discrepancy in discussion section.

Previous reports already showed uNGAL was good biomarker for AKI. In this manuscript, authors accepted ACR criteria for renal response using only reduction level of proteinuria. According to the criteria, there was another response criterion based on eGFR change. I suggest authors to create another table comparing baseline clinical features including eGFR in patients with CR, PR and NR. Cumulative CR or PR rate using criterion based on eGFR change also needs to be shown.

Minor

ROC analysis for predicting proliferative LN using 2-urinary biomarkers needs to be done before using 3 (page15, line 226).

Reviewer #2: The authors implemented a retrospective study to show the utility of urinary biomarkers in predicting renal pathology and treatment response after induction therapy in Chinese patients with SLE who presented lupus nephritis (LN). Receiver operating characteristics (ROC) curve analysis showed a difference in biomarkers between proliferative LN and membranous LN. The results are interesting and support the utility of urinary biomarkers in LN patients.

However, it is difficult to accept the author's opinion since this manuscript contains serious problems such as imprecision due to the extremely small number of cases, lack of induction therapy for LN according to recommendations by the American College of Radiology (ACR), the European League Against Rheumatism (EULAR), or the Kidney Disease Improving Global Outcomes (KDIGO) in this patient population, and the lack of information regarding levels of conventional laboratory measures and histologic features in the studied subjects.

Specific comments:

1. The objective of this analysis was to predict renal pathology in lupus nephritis of SLE patients using urinary biomarkers. The biomarkers shown by Liu L have already been demonstrated to be potential predictors of renal pathology feature not only in lupus nephritis but also in glomerulonephritis including IgA nephropathy and diabetic nephropathy. The current study by Liu requires further optimization and validation in another cohort.

2. Table 1: The authors should show more information regarding levels of conventional laboratory measures and histologic features in the studied subjects.

3. Materials and methods, Disease assessment and follow-up: As for the assessment of urinary protein levels, the urinary protein ranges overlap between the active LN group (0.5g/24h) and the partial response group (urinary protein creatinine ratio: 0.2-2.0). Evaluation of urinary protein in patients with LN should be assessed by decrease in urinary protein levels over time.

6. PLOS authors have the option to publish the peer review history of their article (what does this mean?). If published, this will include your full peer review and any attached files.

Reviewer #1: No

Reviewer #2: No

---

## [Author Response · Author response to Decision Letter 0]

3 Oct 2020

Responses to reviewers’ comments 

We appreciate the reviewers’ efforts in assessing our manuscript and giving constructive criticism. Base on the reviewers’ comments, we have made major revision to the original manuscript. We believe that the manuscript has been further improved.

Reviewer’s Comments

Reviewer #1: Liu L et al investigated a large number of SLE patients to identify the predictor for renal pathological activity and prognosis by using 10-urinary biomarkers. This study is focusing on the important issue estimating activity of lupus nephritis without pathological confirmation. This manuscript sounds interesting, but I have concerns which authors should address.

Major

Urinary NGAL was highly correlated with both 24-hours proteinuria and RSLEDAI comparing to other urinary biomarkers (Table 3) and only the urinary biomarker predicting renal response. However, it could not distinguish proliferative LN from membranous LN while uVCAM-1, uCystatinC, and uKim-1 were significantly increased in proliferative LN (Figure 2). Author needs to discuss this discrepancy in discussion section.

Re: We have added the following discussion in the manuscript: 

Although urinary NGAL level correlated well with 24h urine protein and predicted renal disease activity, it couldn’t differentiate proliferative LN from membranous LN. Previous studies also haven’t validated the correlation ship between urinary NGAL and the histopathological type of LN. Since patients’ baseline character including renal pathological characteristics were not predictors for treatment response [40, 41], it’s not surprising that urinary NGAL could predict treatment response but not renal histopathology type while the biomarkers which could differentiate different pathological types of LN could not predict treatment response. 

Previous reports already showed uNGAL was good biomarker for AKI. In this manuscript, authors accepted ACR criteria for renal response using only reduction level of proteinuria. According to the criteria, there was another response criterion based on eGFR change. I suggest authors to create another table comparing baseline clinical features including eGFR in patients with CR, PR and NR. Cumulative CR or PR rate using criterion based on eGFR change also needs to be shown.

Re: We appreciate the insightful suggestion. We fully agree with the concern that using only proteinuria level as renal response might bias the results. We apologize for not describing the ACR criteria for renal response precisely in the manuscript. It actually included the requirement of eGFR change. It has now been corrected in the manuscript as follows: 

Complete response (CR): eGFR of >90 ml/minute/1.73m2, urinary protein creatinine ratio (UPCR) <0.2. Partial response (PR): no more than 25% decline in estimated GFR or end-stage renal disease, at least 50% reduction in the UPCR and UPCR is 0.2–2.0. Non-response (NR): not meeting the CR or PR criteria. Based on eGFR and UPCR, the patients were divided into CR, PR and NR groups. We didn’t include urine sediment in the criteria of treatment response due to the poor reproductivity. 

We have double checked our original data and confirmed that the renal response status was defined base on both proteinuria and eGFR change.

We have created another table comparing baseline clinical features of the patients with CR, PR and NR as S1 Table. We also did ANOVA test to check if baseline eGFR levels differed in CR, PR, and NR patients. It turned out that the baseline eGFR levels were not significantly different among the three groups (see the following figure 1’). We didn’t calculate the cumulative CR or PR rate using criterion based on eGFR change due to the following two reasons. Firstly, using only eGFR change to define the treatment response in lupus nephritis is not widely accepted. Secondly, the calculation of cumulative CR or PR rate was not doable since the we only have detailed laboratory data at the 6 months.

Figure 1’. Baseline eGFR levels in patients with different treatment responses. (shown in the attached file)

Minor

ROC analysis for predicting proliferative LN using 2-urinary biomarkers needs to be done before using 3 (page15, line 226).

Re: Thanks for the suggestion. We have replotted the ROC curve in Figure 2(D), in which we have added ROC curve for predicting proliferative LN using 2-urinary biomarkers. 

Reviewer #2: The authors implemented a retrospective study to show the utility of urinary biomarkers in predicting renal pathology and treatment response after induction therapy in Chinese patients with SLE who presented lupus nephritis (LN). Receiver operating characteristics (ROC) curve analysis showed a difference in biomarkers between proliferative LN and membranous LN. The results are interesting and support the utility of urinary biomarkers in LN patients.

However, it is difficult to accept the author's opinion since this manuscript contains serious problems such as imprecision due to the extremely small number of cases, lack of induction therapy for LN according to recommendations by the American College of Radiology (ACR), the European League Against Rheumatism (EULAR), or the Kidney Disease Improving Global Outcomes (KDIGO) in this patient population, and the lack of information regarding levels of conventional laboratory measures and histologic features in the studied subjects.

Specific comments:

1. The objective of this analysis was to predict renal pathology in lupus nephritis of SLE patients using urinary biomarkers. The biomarkers shown by Liu L have already been demonstrated to be potential predictors of renal pathology feature not only in lupus nephritis but also in glomerulonephritis including IgA nephropathy and diabetic nephropathy. The current study by Liu requires further optimization and validation in another cohort.

Re: We fully understand the concern and totally agree with the idea that the biomarkers validated in this study were not specific for LN. We envision the clinical application of these biomarkers in predicting renal pathology features of LN in patients who have already had the diagnosis of SLE base on the classification criteria. In this context, it is less likely that the patients will have another type of nephropathy. However, we do agree that future studies should be designed to include patients with different types of nephropathy to clarify if the urinary biomarkers validated in this study were specific to LN or not. 

To clarify this, we have added the following discussion in the manuscript:

Most of the biomarkers validated in this study, however, are not specific to LN. For instants, urinary KIM-1 and NGAL has been reported as biomarkers of kidney injury in obese children [42]. The current study didn’t include patients with other chronic kidney diseases. We envision the clinical implication of the current study is to use these biomarkers in predicting renal pathology features or treatment response of LN after patients’ diagnosis of SLE base on the classification criteria.

2. Table 1: The authors should show more information regarding levels of conventional laboratory measures and histologic features in the studied subjects.

Re: Thanks for the suggestion. We have added more detailed clinical and laboratory data and significantly improved Table 1. Unfortunately, we don’t have more detailed information regarding histological features except for historical type, activity index, and chronicity index of the active LN patients.

3. Materials and methods, Disease assessment and follow-up: As for the assessment of urinary protein levels, the urinary protein ranges overlap between the active LN group (0.5g/24h) and the partial response group (urinary protein creatinine ratio: 0.2-2.0). Evaluation of urinary protein in patients with LN should be assessed by decrease in urinary protein levels over time.

Re: We fully agree with the concern and we apologize for not describing our criteria on the treatment response precisely in the manuscript. It has now been corrected as follows in the manuscript:

Complete response (CR): eGFR of >90 ml/minute/1.73m2, urinary protein creatinine ratio (UPCR) <0.2. Partial response (PR): no more than 25% decline in estimated GFR or end-stage renal disease, at least 50% reduction in the UPCR and UPCR is 0.2–2.0. Non-response (NR): not meeting the CR or PR criteria. Based on eGFR and UPCR, the patients were divided into CR, PR and NR groups. We didn’t include urine sediment in the criteria of treatment response due to the poor reproductivity.

We have also double checked our patients’ response status based on the above criteria and confirmed that the renal response status was defined base on both absolute value of proteinuria as well as improvement of proteinuria, and eGFR change.

---

## [Decision Letter · Decision Letter 1]

6 Oct 2020

The utility of urinary biomarker panel in predicting renal pathology and treatment response in Chinese lupus nephritis patients

PONE-D-20-15842R1

Dear Dr. Bao,

We’re pleased to inform you that your manuscript has been judged scientifically suitable for publication and will be formally accepted for publication once it meets all outstanding technical requirements.

Kind regards,

Masataka Kuwana, MD, PhD

Academic Editor

PLOS ONE

Additional Editor Comments (optional):

Reviewers' comments:

Reviewer's Responses to Questions

**Comments to the Author**

1. If the authors have adequately addressed your comments raised in a previous round of review and you feel that this manuscript is now acceptable for publication, you may indicate that here to bypass the “Comments to the Author” section, enter your conflict of interest statement in the “Confidential to Editor” section, and submit your "Accept" recommendation.

Reviewer #1: All comments have been addressed

2. Is the manuscript technically sound, and do the data support the conclusions?

Reviewer #1: Yes

3. Has the statistical analysis been performed appropriately and rigorously? 

Reviewer #1: Yes

4. Have the authors made all data underlying the findings in their manuscript fully available?

Reviewer #1: Yes

5. Is the manuscript presented in an intelligible fashion and written in standard English?

Reviewer #1: Yes

6. Review Comments to the Author

Reviewer #1: (No Response)

7. PLOS authors have the option to publish the peer review history of their article (what does this mean?). If published, this will include your full peer review and any attached files.

Reviewer #1: No

---

## [Editor Report · Acceptance letter]

19 Oct 2020

PONE-D-20-15842R1 

The utility of urinary biomarker panel in predicting renal pathology and treatment response in Chinese lupus nephritis patients 

Dear Dr. Bao:

I'm pleased to inform you that your manuscript has been deemed suitable for publication in PLOS ONE. Congratulations! Your manuscript is now with our production department. 

Kind regards, 

on behalf of

Prof. Masataka Kuwana 

Academic Editor

PLOS ONE